# The Role of miR-486-5p on CSCs Phenotypes in Colorectal Cancer

**DOI:** 10.3390/cancers16244237

**Published:** 2024-12-19

**Authors:** Federica Etzi, Carmen Griñán-Lisón, Grazia Fenu, Aitor González-Titos, Andrea Pisano, Cristiano Farace, Angela Sabalic, Manuel Picon-Ruiz, Juan Antonio Marchal, Roberto Madeddu

**Affiliations:** 1Department of Biomedical Science, University of Sassari, 07100 Sassari, Italy or f.etzi@studenti.uniss.it (F.E.); g.fenu@studenti.uniss.it (G.F.); cristiano.farace@hotmail.it (C.F.); angelasabalic08@gmail.com (A.S.); rmadeddu@uniss.it (R.M.); 2Department of Biochemistry and Molecular Biology 2, Faculty of Pharmacy, University of Granada, Campus de Cartuja s/n, 18071 Granada, Spain; 3Centre for Genomics and Oncological Research, GENYO, Pfizer/University of Granada/Andalusian Regional Government, 18016 Granada, Spain; 4Instituto de Investigación Biosanitaria ibs.GRANADA, University Hospitals of Granada, University of Granada, 18012 Granada, Spain; aitorgt@ugr.es (A.G.-T.); mpicon@ugr.es (M.P.-R.); jmarchal@ugr.es (J.A.M.); 5Excellence Research Unit “Modeling Nature” (MNat), University of Granada, 18100 Granada, Spain; 6Biopathology and Regenerative Medicine Institute (IBIMER), Centre for Biomedical Research (CIBM), University of Granada, 18100 Granada, Spain; 7Department of Human Anatomy and Embryology, Faculty of Medicine, University of Granada, 18016 Granada, Spain; 8National Institute of Biostructures and Biosystems, 00136 Rome, Italy

**Keywords:** colorectal cancer, spheres, cancer stem cells, miR-486-5p, oncosuppressive

## Abstract

**Simple Summary:**

Previous studies have indicated that the presence of cancer stem cells may be a contributing factor to the development of metastasis in colorectal cancer patients. Cancer stem cells represent a small subpopulation within the tumor mass that exhibits heightened resistance to treatment and possesses the capacity for self-replication, epithelial–mesenchymal transition, and the generation of new tumors. The tumor microenvironment secretes and releases several molecules that facilitate the self-renewal of cancer stem cells and provide support for colorectal cancer progression. microRNAs are involved in direct cell-to-cell signaling and paracrine signaling between tumor cells and other tumor microenvironment components. They could act as tumor suppressors or oncomiRs, and their deregulation is involved in colorectal cancer progression and cancer stem cell formation. In our previous studies, we demonstrated the oncosuppressive function of miR-486-5p in colorectal cancer; these findings prompted us to conduct a more detailed investigation into its role in cancer stem cell phenotypes.

**Abstract:**

Background: Colorectal cancer (CRC) is the third diagnosed cancer worldwide. Forty-four percent of metastatic colorectal cancer patients were diagnosed at an early stage. Despite curative resection, approximately 40% of patients will develop metastases within a few years. Previous studies indicate the presence of cancer stem cells (CSCs) and their contribution to CRC progression and metastasis. miRNAs deregulation plays a role in CSCs formation and in tumor development. In light of previous studies, we investigated the role of miR-486-5p to understand its role in CSC better. Methods: The expression of miR-486-5p was assessed in adherent cells and spheres generated from two CRC cell lines to observe the difference in expression in CSC-enriched spheroids. Afterward, we overexpressed and underexpressed this miRNA in adherent and sphere cultures through the transfection of a miR-486-5p mimic and a mimic inhibitor. Results: The results demonstrated that miR-486-5p exhibited a notable downregulation in CSC models, and its overexpression led to a significant decrease in colony size. Conclusions: In this study, we confirmed that miR-486-5p plays an oncosuppressive role in CRC, thereby advancing our understanding of the role of this microRNA in the CSC phenotype.

## 1. Introduction

Colorectal cancer (CRC) is the third most frequently diagnosed cancer worldwide, with an annual incidence of 10.7%, after breast cancer (11.7%) and lung cancer (11.4%), and it is the second leading cause of cancer death, with 9.5% mortality, second to lung cancer [1]. According to the most recent data, about 10% of survivors live with metastatic cancer, 44% of whom were diagnosed at an early stage [2]. The disease, which is quite rare before the age of 40, is most common in people between 60 and 75; both incidence and mortality rates are higher in males than females, with an age-standardized rate (ASR) incidence of 23.4 for men and 16.2 for women and an ASR mortality of 11.1 for men and 7.2 for women (ASR per 100,000 worldwide) [1]. It is estimated that approximately 30–40% of CRC patients who undergo curative resection of the primary tumor will develop metastases within a few years and that most of these recurrences will occur within the first two years [2,3]. In recent years, screening techniques have made it possible to lower the age of diagnosis when the cancer is in the early stages; in fact, the overall incidence of CRC decreased in individuals over 50 years old but increased in those under 50 years old [4]. As previously mentioned, the 5-year survival of CRC patients varied depending on the tumor stage at the time of diagnosis. Currently, colonoscopy is recognized as the gold standard for the detection of CRC, but it has limited application because of its invasiveness, time-consuming nature, high cost, and high operator variability [2]. Common screening methods, such as the fecal occult blood test (FOBT) and prognostic tests, such as carcinoembryonic antigen (CEA) quantification, microsatellite instability assessment, and mutations in the most frequently mutated genes KRAS (proto-oncogene KRAS, GTPase), NRAS (proto-oncogene NRAS, GTPase), BRAF (v-RAF murine sarcoma viral oncogene homolog B), and mismatch repair (MMR), unfortunately, show low sensitivity [2]. The genetic and epigenetic events involved in adenoma to carcinoma transition and CRC progression are supported by the tumor microenvironment (TME) [5]. The TME represents a very complex network between tumor cells and stromal, endothelial, and immune cells, which contributes to the determination of an aggressive tumor phenotype; the presence of inflammatory cells and inflammatory mediators such as chemokines and cytokines facilitates tumor progression, including CRC, by maintaining paracrine signaling between tumor-resident adipocytes, that provide a rich source of energy, and tumor cells, that require energy for their high proliferation [5,6]. The majority of cells within the tumor mass lack self-renewal capacity and are not tumorigenic. However, within the wider TME, cancer stem cell (CSC) niches are anatomically diverse microenvironments in which cells secrete molecules that encourage self-renewal of CSCs, cause angiogenesis and attract immune cells and other factor-secreting stromal cells [7]. CSCs are a small subpopulation of cells in the tumor mass that are immortalized, possess the capacity for self-renewal, asymmetrical self-renewal, pluripotency, the ability to restart the original tumor, and are involved in tumor growth, initiation, maintenance, survival, metastasis, cancer recurrence, and increased aldehyde dehydrogenase 1 enzymatic activity [8]. In spite of their fundamental role in cancer, CSCs represent only 0.1–10% of the cells present in it. The property of pluripotency and asymmetric cell division allows CSCs to generate heterogeneous lineages of tumor cells with different phenotypes, resulting in the growth of the primary tumor and the insurgence of new tumors [9]. The presence of colorectal CSCs (CCSCs) or colon-cancer-initiating cells was first proven in 2007 by Ricci-Vitiani et al. [10], O’Brien et al. [11], and Delerba et al. [12]. The origin of CCSCs is still debated; numerous evidence suggests that CCSCs can be generated from intestinal stem cells (ISCs) or from differentiated intestinal cells that can acquire stem-like characteristics and become CCSCs through a process of dedifferentiation. Genetic, epigenetic, and even niche and microenvironmental conditions contribute to this transformation [7,13,14]. Physiologically, the ISC compartment maintains tissue homeostasis by generating new cells that ascend the crypt as differentiated cells and eventually replace apoptotic cells at the top of the crypt [9,14]. The rapid and continuous regeneration to which the intestinal epithelium is subjected, supported by cryptic ISCs, greatly increases the risk of malignant conversion [13].

MicroRNAs (miRNAs) are endogenous, short, noncoding RNA sequences of 18–25 nucleotides that regulate gene expression at the post-transcriptional level in a sequence-specific manner [15]. Due to their high stability and the possibility of detecting them in human body fluids, miRNAs are being studied as a new class of valuable biomarkers. Numerous studies have demonstrated their utility in biomedical research, particularly in cancer [16]. For instance, miR-128-3p has shown a key role in suppressing breast cancer cellular progression by targeting LIMK1, highlighting its potential as a biomarker and therapeutic target in this disease [17]. Similarly, the rs12976445 polymorphism in miR-125a has been identified as a factor altering its expression, influencing breast cancer susceptibility and progression, further reinforcing the importance of miRNAs as molecular markers [18]. Indeed, increasing evidence indicates that deregulated expression of miRNAs plays a functional role in CRC, acting as tumor suppressors or oncogenes to regulate the expression of their specific target mRNAs [19]. miRNAs have a significant role in maintaining the physiology of normal colon cells, while alteration of their levels contributes to CRC development, progression, and metastasis, drug resistance, tumor recurrence, and plays a role in CSC formation and epithelial-to-mesenchymal transition (EMT) [19]. In addition, miRNAs are involved in direct cell-to-cell signaling and paracrine signaling between tumor cells and other TME components as secreted molecules in exosomes or microvesicles [20]. In previous studies, we confirmed the role of miRNAs in CRC by investigating the expression of a set of miRNAs selected from literature in CRC patients’ cancer tissue, healthy tissue, and serum and determining the relationships with their clinical parameters. Additionally, we investigated which miRNAs are associated with CSC phenotype using two different CSC in vitro models obtained from human-established CRC cell lines [21]. The miR-486-5p has been observed to exhibit altered expression in various tumors [22,23]. It is considered an onco-suppressor in CRC due to its gradual downregulation in tissues as the pathology advances [24]. In our previous research, we identified miR-486-5p as a promising candidate for CRC biomarker studies based on its differential expression in CRC patient samples (serum and stool) across tumor stages, along with our meta-analysis using GEO datasets. Our findings showed that miR-486-5p was downregulated in metastatic patients and CSC models, compared to healthy controls and parental cells. miRNA’s stability in biological fluids, coupled with its deregulation in CRC, makes it a promising candidate for diagnostic and prognostic applications in CRC, motivating our continued investigation [25].

In this study, we aimed to explore the role of miR-486-5p in CRC and CSC phenotypes. Using colonospheres from colon cancer cell lines, we analyzed the presence of CSCs through increased expression of EMT and stemness markers. By manipulating miR-486-5p levels via mimic and mimic inhibitor transfection, we studied the functional impact of miR-486-5p overexpression and downregulation in colonospheres derived from two human CRC cell lines.

## 2. Materials and Methods

### 2.1. Cell Lines and Cell Culture

Two cancer cell lines, HT-29 (ATCC HTB-38, representative of the primary tumor, with mutations in APC, BRAF, PIK3CA, SMAD4, and TP53 genes) and T84 (ATCC CCL-248, representative of metastatic cancer, with mutations in APC, KRAS, PIK3CA, and TP53 genes) were purchased from ATCC and were employed to set up two culture models: in adherent conditions and nonadherent conditions (spheres or colonospheres).

We chose these two cell lines because they represent distinct stages of colorectal cancer progression. HT29 models a well-differentiated primary colorectal adenocarcinoma with high proliferative capacity and morphology closely resembling the original tumor. In contrast, T84, derived from liver metastases, serves as a representative model of metastatic colorectal cancer, enabling the study of tumor progression mechanisms. These lines may exhibit differences in microRNA expression patterns, allowing comparisons between primary and metastatic tumors. For example, microRNAs associated with metastasis or chemoresistance might be more expressed in T84. The use of these complementary models allows for the investigation of diagnostic or therapeutic markers in a context that mirrors the clinical progression of the disease. In adherent culture conditions, both cell lines were cultured following American Type Culture Collection (ATCC; Manassas, VA, USA) recommendations in supplemented Dulbecco’s modified Eagle’s medium (DMEM) containing 10% fetal bovine serum (FBS) and 1% penicillin/streptomycin (Pen-Str P-0781; Sigma, St. Louis, MO, USA) at 37 °C in 5% CO_2_. In nonadherent cell culture models, secondary spheres enriched in CSCs were obtained from HT-29 and T84 cell lines following the patented protocol WO2016020572A1 [8]. The protocol was developed to yield spheres in Corning^®^ Costar^®^ Ultra-Low Attachment Multiwell Plates, utilizing spheres culture medium obtained supplementing the DMEM/F-12 nutrient mixture without FBS with 1% penicillin/streptomycin (Pen-Str P-0781; Sigma-Aldrich, St. Louis, MO, USA), 1× B-27 (B-27™ Supplement [50×], Minus Vitamin A; Invitrogen, Waltham, MA, USA), 10 µg/mL insulin (Insulin–Transferrin–Selenium [ITS-G, 100×]; Invitrogen, Waltham, MA, USA), 4 ng/mL heparin (cell-culture-tested heparin sodium; Sigma, St. Louis, MO, USA), 1 µg/mL hydrocortisone (Sigma-Aldrich, St. Louis, MO, USA), 10 ng/mL epidermal growth factor (Sigma-Aldrich, St. Louis, MO, USA), 10 ng/mL interleukin 6 (Miltenyi, Bergisch Gladbach, Germany), 10 ng/mL fibroblast growth factor (Sigma), and 10 ng/mL hepatocellular growth factor (Miltenyi, Bergisch Gladbach, Germany). After 72 h of incubation at 37 °C in 5% CO_2_, primary spheres were obtained. They were collected by centrifugation, disaggregated with trypsin-EDTA, and mechanically disrupted with a pipette. The trypsin was inactivated with the addition of DMEM containing serum, and cells were washed with phosphate-buffered saline (PBS) to remove FBS’s traces. Single cells were then resuspended in spheres culture medium and plated in ultra-low adherence multiwell plates. Following an additional incubation period of 72 h at 37 °C in 5% CO_2_, secondary spheres were generated.

### 2.2. Transient Transfection with Synthetic miRNA-486-5p Mimics and Inhibitors

The upregulation and the inhibition of miR-486-5p were induced in monolayer cells and spheres by the transfection of miR-486-5p mimic or inhibitor (Qiagen, Hilden, Germany), respectively, paired with the relative controls. The 5′-FAM-fluorescence-labelled delivery control (Qiagen, Hilden, Germany) was used to measure the transfection efficiency in HT-29 and T84 monolayer cells and colonospheres. The TransIT-X2^®^ Transfection Reagent (Mirus Bio, Madison, WI, USA) was used according to the manufacturer’s instructions to perform the transfection. The miRNA mimic and inhibitor were used at the final concentration of 5 nM and 50 nM, respectively and were prepared in 3 µL/mL of TransIT-X2^®^ and 100 µL/mL of Opti-MEM medium (Gibco, New York, NY, USA) into a well of standard 24-well plates containing 6 × 10^3^ cells in 0.4 mL of medium [25]. The prepared reagents were allowed to stand at room temperature for 15 to 30 min and then added directly to the cell culture medium. Prior to further analysis, the cells were cultured for 3 days at 37 °C in a 5% CO_2_ atmosphere.

### 2.3. RNA Extraction from Cells

Monolayer cells (HT-29 and T84) and derived colonospheres were harvested and disaggregated with trypsin-EDTA, pelleted with centrifugation at 1500× *g* for 5 min, and washed twice with PBS. The total RNA was extracted by adding 1 mL of TRI Reagent (Sigma-Aldrich, St. Louis, MO, USA) from the pellet. After 15 min of incubation at room temperature, 200 µL of chloroform was added, and the cells were vortexed and allowed to stand at room temperature for 10 min. Then, the cells were centrifuged for 10 min at 12,000× *g* at 4 °C, and the supernatant was transferred into a new microtube. After, 500 µL of isopropanol was added to the supernatant, which was vortexed, incubated for 10 min at room temperature and centrifuged for 10 min at 12,000× *g* at 4 °C. The supernatant was eliminated and replaced with a 75% ethanol solution. The sample was centrifuged for 5 min at 17,000× *g* at 4 °C. Then, the supernatant was eliminated, and the sample was allowed to dry at room temperature for 1 h. After the pellet was resuspended in 50 µL of Milli-Q™ water, the RNA concentration and quality were evaluated with a NanoDrop spectrophotometer (Thermo Fisher Scientific, Waltham, MA, USA).

### 2.4. Retrotranscription and Real-Time PCR for miRNA Expression

To obtain the cDNA from extracted RNAs, each sample was diluted in nuclease-free water to achieve a final concentration of 5 ng/µL of RNA. The miRCURY™ LNA™ RT Kit (Qiagen, Hilden, Germany) was used according to the manufacturer’s instructions to synthesize cDNA, and the thermocycler was programmed in accordance with the specified protocol: 60 min at 42 °C, 5 min at 95 °C, cooling to 4 °C. The reactions were spiked with exogenous UniSp6 RNA (RNA Spike-In Kit, Qiagen, Hilden, Germany). The samples were then stored at −20 °C until processing. Quantitative real-time PCR (quantitative PCR or qPCR) was performed with miRCURY LNA SYBR^®^ Green PCR Kit (Qiagen, Hilden, Germany). miRCURY LNA miRNA PCR primers (Qiagen, Hilden, Germany) were used for miRNA-486-5p (hsa-miR-486-5p, Qiagen, Hilden, Germany). The U6 snRNA housekeeping gene was used for data normalization and the UniSp6 primer set was assessed for cDNA quality. For the real time-PCR assays, the cDNA was diluted 1:80 in nuclease-free water, and 4 µL of diluted cDNA was mixed with 5 µL of PCR master mix, 1 µL of nuclease-free water, and 1 µL of each primer. The cDNAs were amplified using the StepOne™ Real-Time PCR system (Applied Biosystems™, Waltham, MA, USA), which was configured as follows: the first cycle was conducted at 95 °C for 10 min, followed by 45 cycles in which a 95 °C cycle was maintained for 10 s and a 60 °C cycle was maintained for 1 min, with a ramp rate of 1.6 °C/s. Relative quantification of miRNA expression was calculated using the 2^−∆∆Cq^ method and each reaction was performed in triplicate.

### 2.5. Retrotranscription and Real-Time PCR Assay for Stemness and EMT Genes Expression

The total RNA extracted, as previously described in Section 2.3, was retrotranscripted with the GoScript™ Reverse Transcription System (Promega, Madison, WI, USA), according to the manufacturer’s instructions. One µg of each extracted RNA was diluted with 9 µL of nuclease-free water, heated for 10 min at 70 °C, and kept on ice until the reverse transcription reaction mix was added. The thermal cycler was set up according to the protocol: 60 min at 42 °C, 5 min at 95 °C, cooling to 4° C. The qPCR was performed using GoTaq^®^ qPCR Master Mix (Promega, Madison, WI, USA) and primers from the StemElite™ (Promega, Madison, WI, USA). The following primers were used to assess the expression of four stemness genes: *SOX-2* (forward sequence: 5′-GGAAAGTTGGGATCGAACAA-3′; reverse sequence 5′-GGAGCTTTGCAGGAAGTTTG-3′), *KLF-4* (forward sequence: 5′-CGAACCCACACAGGTGAGAA-3′; reverse sequence: 5′-TACGGTAGTGCCTGGTCAGTTC-3′), *c-Myc* (forward sequence: 5′-CTTTTCCTGTCCTGTCCCAAC-3′; reverse sequence: 5′-CTGCTTTACGCTCAT-3′) and *OCT-4* (forward sequence: 5′-TCTCGCCCCCTCCAGGT-3′; reverse sequence: 5′-GCCCCACTCCAACCTGG-3′). The following primers were used to assess the expression of three EMT genes: *Vimentin* (forward sequence: 5′-GAACCTGAGGGAAACTAATC-3′; reverse sequence: 5′-GAAAGGCACTTGAAAGCT-3′); *SLUG* (forward sequence: 5′-TGGTTGCTTCAAGGACACAT-3′; reverse sequence: 5′- GTTGCAGTGAGGGCAAGAA-3′) and *SNAIL* (forward sequence: 5′-TACAAAAACCCACGCAGACA-3′; reverse sequence: 5′-ACCCCACATCCTTCTCACTG-3′). *GAPDH* was used as a housekeeping gene (forward sequence: 5′-CAACAATATCCACTTTACCAGAG-3′; reverse sequence: 5′-TCGGAGTCAACGGATTTG-3′). The cDNA was diluted 1:80 in nuclease-free water, and 1 µL of the diluted solution was added to the Master Mix prepared in accordance with the kit protocol. The cDNAs were amplified using the StepOne™ Real-Time PCR system (Applied Biosystems™, Waltham, MA, USA) configured as follows: the first cycle was conducted at 90 °C for 2 min, followed by 40 cycles in which a 95 °C cycle was maintained for 15 s and a 60 °C cycle was maintained for 1 min, with a ramp rate of 1.6 °C/s. Relative quantification of miRNA expression was calculated using the 2^−∆∆Cq^ method. Each reaction was performed in triplicate on monolayer cells and spheres prior to and following the transfection of miR-486-5p mimic and inhibitor.

### 2.6. Cell Viability Assay

The Alamar Blue assay (alamarBlue™ Cell Viability Reagent, Invitrogen, Waltham, MA, USA) was performed to evaluate the cell viability before and after the miRNA-486-5p transfection in monolayer cells and spheres obtained from HT-29 and T84 cell lines. Cells were seeded in a 96-well plate in 100 µL of complete medium at the concentration of 3000 cells/well. Once cell adhesion had occurred, 10 µL of alamarBlue reagent was added to 90 µL of culture medium in each well, and the plates were incubated at 37 °C in a 5% CO_2_ atmosphere for one hour. The fluorescence intensity was measured using a plate reader (Synergy HT, BIO-TEK, Winooski, VT, USA). The data were normalized, and the viability rate was calculated.

### 2.7. Flow Cytometry

The ALDEFLUOR kit (Stem Cell Technologies, Vancouver, BC, Canada) was performed according to the manufacturer’s instructions to detect the ALDH1 activity in adherent cells and secondary spheres of HT-29 and T84 cell lines before and after the transfection of miR-486-5p mimic and inhibitor. The diethylaminobenzaldehyde (DEAB) was used as an ALDH1 inhibitor to set ALDH1 gates. Cells were harvested in three tubes per treatment and centrifuged at 250× *g* at 4 °C for 5 min. The supernatant was removed and the cell pellet was resuspended in 500 μL of ALDEFLUOR Buffer. One of the three tubes was treated with 5 µL of ALDEFLUOR reagent (ALDH1), while the second tube was treated with 5 µL of ALDEFLUOR reagent and 10 µL of DEAB reagent. The third tube was used as a control negative. All tubes were incubated at 37 °C for 30 min and then centrifuged at 250× *g* at 4 °C for 5 min. Once the supernatant had been removed, the pellet was resuspended in 400 µL of ALDEFLUOR Buffer for the subsequent flow cytometry reading.

### 2.8. Soft Agar Colony Formation Assay

The soft agar colony formation assay was employed, as previously described by our group [26], to assess the clonogenic activity of colonospheres derived from the two cell lines (HT-29 and T84) prior to and following the transfection of miR-486-5p mimic and inhibitor. The bottom of a 24-well ultra-low attachment plate cell culture was prepared as follows: 500 µL of 0.8% agar solution in supplemented DMEM was prewarmed to 37 °C and dispensed into each well avoiding bubble formation on the surface and allowed to solidify at room temperature. Secondary spheres were harvested and disaggregated using trypsin-EDTA. The top layer solution was prepared by adding 2 × 10^4^ cells/mL in 0.4% agar solution in supplemented DMEM prewarmed to 37 °C. A total of 500 µL of the top layer solution was dispensed in each well, avoiding bubble formation on the surface and allowing it to solidify at room temperature to obtain a final concentration of 104 cells per well. After solidification, 200 µL of supplemented DMEM was added to each well as a feeder layer and replaced every 3–4 days to feed the cells. Transfection reagents (miR-486-5p mimic and inhibitor prepared as previously described in Section 2.2) were added to the feeder layer and replaced every 3–4 days. Cells were cultured at 37 °C in 5% CO_2_ for 20 days. Then, cells were stained with 500 μL of iodonitrotetrazolium chloride (Sigma-Aldrich, St. Louis, MO, USA) solution (10 g of iodonitrotetrazolium chloride dissolved in 10 mL of sterile H_2_O) and incubated at 37 °C in 5% CO_2_ for 24 h. Wells were then washed with PBS 1× and colonies were counted and analyzed using a dissecting microscope and the ImageJ software (version number 1.54d).

### 2.9. Statistical Analysis

All graphed data are the result of at least three experiments and are presented as the mean ± standard error. Statistical difference was determined from two-tailed Student’s *t*-tests. The values of *p* < 0.05 were deemed to be statistically significant.

## 3. Results

### 3.1. Spheroids and CSC Enrichment in Colorectal Cell Lines HT-29 and T84

Both cell lines were capable of forming colonospheres under serum-free, nonadherent conditions. The secondary spheres derived from HT-29 cell lines were compact and well-shaped (Figure 1a); the T84 cell line formed less compact secondary spheres with a grape-cluster shape (Figure 1b). The evaluation of CSC marker ALDH1 expression confirmed the presence of CSCs in the sphere culture models. The percentage of ALDH1-positive cells increased significantly from 6.1% in HT-29 adherent cells to 74.2% in HT-29 colonospheres (Figure 1c) and from 10.85% in T84 in monolayer cells to 24.55% in T84 colonospheres (Figure 1d). Cell viability was evaluated in monolayer and CSC culture models of HT-29 and T84 cell lines at time 0 (T0) and after 72 h of incubation (T72) with Alamar blue assay. The HT-29 cell line exhibited a significantly higher viability rate in the CSC culture than in the monolayer culture (*p* < 0.001) (Figure 1e). In contrast, the T84 cell line showed a lower viability rate in spheroids than in the monolayer counterpart (Figure 1f).

### 3.2. Expression Levels of EMT and Stemness Genes Change Between Adherent Cells and Colonospheres

The expression of three EMT genes (*SNAIL*, *SLUG*, and *Vimentin*) and four stemness genes (*OCT-4*, *c-Myc*, *SOX-2*, and *KLF-4*) were evaluated by qPCR in monolayer and CSCs culture. The HT-29 CSCs exhibited reduced levels of *SNAIL* and *SLUG* expression compared with parental adherent cells with a fold change of 0.16 (*p* < 0.001) and 0.46 (*p* < 0.05), respectively, and higher levels of *Vimentin* in CSCs compared with monolayer-cultured cells with a fold change of 71.30 (*p* < 0.05). The expression of stemness genes *OCT-4* and *SOX-2* was significantly higher in spheres derived from the HT-29 cell line with a fold change of 2.99 (*p* < 0.01) and 2.11 (*p* < 0.01), respectively, but *c-Myc* and *KLF-4* genes were downregulated in CSCs with a fold change of 0.27 (*p* < 0.001) and 0.22 (*p* < 0.001). (Figure 2a). In comparison to adherent cultured cells, T84 CSCs showed an incremented expression of *SLUG* and *Vimentin* genes, with a fold change of 9.46 (*p* < 0.01) and 7.86 (*p* < 0.05), respectively, while *SNAIL* expression remained unaltered. In T84 CSCs, all stemness genes were found to be upregulated in comparison with monolayer cells with a fold change of 29.96 for *OCT-4* (*p* < 0.01), 4.43 (*p* < 0.05) for *c-Myc*, 7.84 (*p* < 0.05) for *SOX-2*, and 3.02 (*p* < 0.05) for *KLF-4* (Figure 2b).

### 3.3. miR-486-5p Is Downregulated in Colonospheres Culture Model

The expression of miR-486-5p was evaluated both in the monolayer culture model and in the CSC culture model derived from HT-29 and T84 cell lines using qPCR. Results show that the expression levels of miR-486-5p were significantly downregulated in the CSC model compared with the monolayer model. Specifically, miR-486-5p was found to be 0.23-fold lower in HT-29 sphere-cultured cells in comparison to the monolayer counterpart (*p* < 0.05) (Figure 3a), and 0.0015-fold lower in T84 spheroid in comparison to the adherent-cultured cells (*p* < 0.01) (Figure 3b).

### 3.4. The Effect of Transfection on Cellular Viability

The cellular viability was assessed by Alamar blue assay at time 0 (T0) and after 72 h (T72) of transfection in monolayer and spheroid cultures of both CRC cell lines. The results were obtained after normalization and expressed as normalized fluorescence. In HT-29 monolayer culture, the viability rate was found to be lower in both mimic (1.35) and inhibitor (1.45) treated cells, with a statistically significant difference observed only in the mimic-treated cells compared with control (nontransfected) cells (1.70) (Figure 4a). In T84 adherent cells, the viability decreased from 8.59 in control cells to 7.08 (*p* < 0.05) in mimic-treated cells and to 7.48 (*p* < 0.05) in inhibitor-treated cells (Figure 4b). The results obtained on the spheres of both cell lines showed no statistically significant differences (Figure 4c,d).

### 3.5. miR-486-5p Module the ALDH1 Activity in Both Adherent and CSCs

The ALDH1 activity was evaluated after miR-486-5p mimic and inhibitor transfection in HT-29 and T84 cell lines. In the HT-29 monolayer, the ALDH1-positive cells increased from 6.1% in control cells to 6.95% (*p* < 0.05) in mimic-treated cells and to 8.75% (*p* < 0.001) in inhibitor-treated cells. A comparison of cells treated with the mimic and cells treated with the inhibitor reveals a statistically significant increase in the percentage of ALDH1-positive cells following treatment with the inhibitor (Figure 5a). In adherence-cultured T84 cells, there were no differences in ALDH1 activity between mimic-treated and control cells. However, there was a significant increase in ALDH1 activity from 10.85% in control cells to 21.5% in inhibitor-treated cells (*p* < 0.001), and a statistically significant increase in inhibitor-treated cells compared with the mimic-treated cells (*p* < 0.001) (Figure 5b). In HT-29 CSCs, the ALDH1 percentage increased from 74.2% in control cells to 83.45% (*p* < 0.001) in mimic-treated CSCs and to 82.7% (*p* < 0.001) in inhibitor-treated cells (Figure 5c). In T84 colonospheres, the percentage of ALDH1-positive cells exhibited a significant decline from 24.55% in nontransfected cells to 13.3% (*p* < 0.001) in mimic-treated cells and to 21.05% (*p* < 0.01) in inhibitor-treated cells. Furthermore, the data indicates a statistically significant increase in the percentage of ALDH1-positive cells following treatment with the inhibitor in comparison to cells treated with the mimic (*p* < 0.01) (Figure 5d).

### 3.6. The Effect of Transfection on Epithelial-Mesenchymal Transition (EMT) and Stemness Genes Expression

The expression levels of EMT and Stemness genes were evaluated in adherent cells, and colonospheres derived from HT-29 and T84 cell lines transfected with miR-486-5p mimic or inhibitor and in nontransfected cells. The levels of *SNAIL* were found to increase following the inhibition of miR-486-5p in HT-29 adherent cells in comparison to the control and mimic-treated cells. Conversely, the levels of *SNAIL* were observed to decrease in inhibitor-treated T84 monolayer cells in comparison to the control and mimic-treated cells. *SLUG* expression was decreased by miRNA inhibition in HT-29 adherent culture compared to control and in HT-29 colonospheres compared to nontransfected colonospheres. In inhibitor-treated T84 colonospheres, it was less expressed with respect to the control, but it resulted in more expression in comparison with mimic-treated cells. The expression levels of *Vimentin* were reduced in HT-29 colonospheres treated with the mimic with respect to control, but it was further reduced compared to the mimic when miRNA was inhibited. *OCT-4* in HT-29 adherent cells was underexpressed after both mimic and inhibitor transfection with respect to untreated cells, but there was a significant increase in its levels in inhibitor-treated cells compared to mimic-treated cells. In HT-29 colonospheres, it was overexpressed after mimic transfection, and it was decreased after inhibition of miR-486-5p with respect to mimic-treated cells in T84 adherent cells and in HT-29 and T84 colonospheres. *c-Myc* expression levels were reduced after both mimic and inhibitor transfection compared to control in T84 adherent cells. Its expression was higher after miRNA inhibition compared to mimic-treated cells in HT-29 monolayer cells and in both colonosphere cultures. Inhibition of miR-486-5p caused a significant reduction of *SOX-2* expression in HT-29 adherent cells and in HT-29 and T84 colonospheres compared to mimic-treated cells. In T84 monolayer cells, it resulted in underexpression after transfection of the mimic with respect to inhibitor transfected cells. Transfection did not appear to affect the expression of *KLF-4* in any of the models employed (Figure 6).

### 3.7. miR-486-5p Inhibition Enhances the Clonogenic Activity in Cells

We tested the in vitro functional characteristic of the miR-486-5p mimic and inhibitor in secondary colonospheres obtained from HT-29 and T84 cell lines in order to reproduce its overexpression or downregulation. We studied their clonogenic activity by colony-formation assay in soft agar. We calculated the mean value of the number of colonies, the mean value of colony size, and the amount of colonies within specific size ranges. Colonies larger than 2500 µm^2^ were counted and analyzed by ImageJ software. The following size ranges were selected for analysis: from 2500 µm^2^ to 5000 µm^2^ µm, from 5100 µm^2^ to 10,000 µm^2^, from 10,100 µm^2^ to 15,000 µm^2^, from 15,100 µm^2^ to 20,000 µm^2^, and from 20,100 µm^2^ and above. miR-486-5p mimic significantly increased the clonogenic activity of HT-29 spheres-derived. Figure 7g graphically illustrates the increase in colony formation (number of colonies) and reduction in colony size observed in mimic-transfected cells relative to non-transfected cells. In contrast, inhibitor-treated cells exhibited a reduction in the number of colonies and an increment in colony size with respect to untreated cells. Specifically, untreated control cells showed an average of 378 colonies with an average size of 21,400 μm^2^; mimic-treated cells showed an average of 616.75 colonies with an average size of 17,279.75 μm^2^; inhibitor-treated cells showed an average of 239.75 colonies with an average size of 37,898.75 μm^2^. Comparing colonies based on size range, it was observed that the percentage of mimic-treated colonies in the 2500–5000 μm^2^ and 5100–10,000 μm^2^ range was higher with respect to inhibitor-treated spheres and to the control cells. The amount of colonies bigger than 20,100 μm^2^ was higher in inhibitor-treated cells with respect to mimic-treated cells and the percentage of mimic-treated colonies bigger than 20,100 μm^2^ was lower with respect to control cells and inhibitor-treated cells (Figure 7). T84 spheres-derived showed a significant increment in the number of colonies in inhibitor-treated spheres with respect to mimic-treated spheres and a significant reduction of colony size in mimic-treated spheres with respect to untreated spheres. Dividing cells on size range, mimic-treated spheres exhibited a higher percentage of colonies between 2500 and 5000 μm^2^ and a lower percentage between 5100 and 10,000 μm^2^ and between 10,100 and 15,000 μm^2^ related to control and inhibitor-treated colonies. The amount of inhibitor-treated colonies in the 2500–5000 μm^2^ range was incremented related to control but reduced with respect to mimic-treated colonies. In the 5100–10,000 μm^2^ and 10,100–15,000 μm^2^ ranges, the percentage of inhibitor-treated colonies was lower with respect to control and higher with respect to mimic-treated colonies. The percentage of inhibitor-treated colonies bigger than 20,100 μm^2^ was lower with respect to the control cells (Figure 8).

## 4. Discussion

Colorectal cancer (CRC) is a well-studied neoplasm with extensive and heterogeneous genomic aberrations, slow progression, and well-defined risk factors [27]. Stage at diagnosis is the most important predictor of survival, but popular screening methods unfortunately show low sensitivity [2]. An increasing body of evidence suggests that the heterogeneity of CRC is related to colorectal cancer stem cells (CCSCs), thereby supporting the hypothesis that the onset, progression and development of drug resistance in CRC may be related to the maintenance of a CCSC phenotype through deregulation of the pathways involved in differentiation, transformation, growth and epithelial-to-mesenchymal transition (EMT) [28]. Recent data have demonstrated the significant role of epigenetics in regulating the function of CRC cells and CCSCs [7]. Noncoding RNAs, particularly miRNAs, regulate gene expression and play key roles in cellular functions like self-renewal and differentiation [29]. Due to their stability in biological samples, miRNAs have emerged as promising biomarkers for CRC [19]. Previous research from our group showed that miR-486-5p was downregulated in the tumor tissue of CRC patients compared with the healthy counterpart, and in serum, it was shown the miR486-5p downregulation in metastatic patients compared to nonmetastatic patients. Including miR-486-5p in predictive models moderately improved diagnostic and prognostic accuracy when combined with other diagnostic and prognostic factors [25]. The miR-486-5p tumor suppressor role has been identified in lung cancer [30], gastric cancer [31], liver cancer [22], renal cancer [32], thyroid cancer [33], and ovarian cancer [34]. In contrast, miR-486-5p was observed to be overexpressed in prostate cancer tissues and cell lines, and in vivo studies have demonstrated that miR-486-5p acts as an oncomiR in prostate cancer, as it plays a pivotal role in prostate cancer pathogenicity [35]. In glioblastoma, miR-486-5p also functions as an oncomiR. Indeed, forced expression of miR-486-5p enhanced the self-renewal capacity of glioblastoma neurospheres, while inhibition of endogenous miR-486-5p activated PTEN and FoxO1 and induced cell death [36]. This dual function can be explained by considering the fact that a single miRNA molecule has the ability to target tens to hundreds of different mRNAs, which may have opposing oncogenic or tumor-suppressive functions [37]. Given the involvement of miRNAs in CSC maintenance, the role of miR-486-5p in the stem cell phenotype has been investigated in CSCs of various tumor types. According to different studies, miR-486-5p may act both as a tumor suppressor and as an oncogene [22,36]. In our previous study, we also evaluated the effects of miR-486-5p on CCSCs using a three-dimensional spheres culture model. The results showed that miR-486-5p has a suppressive role in stemness characteristics, resulting in being downregulated in cancer stem-like cells obtained from different CRC cell lines compared to cells grown in adherent condition, having an effect on stemness genes and the Wnt, Notch, Hedgehog, and TGF-β pathways. Given the results obtained at the molecular level, this study aims to elucidate the phenotypic and functional effects of miR-486-5p [25]. To confirm its role in CRC and to expand our understanding of its involvement in CSC phenotype, in this work, we mimicked an overexpression and a downregulation of the miR-486-5p through transfection of monolayer cells and spheroids enriched in CSCs obtained from the HT-29 and T84 cell lines, representative of the primary colorectal tumor and the metastatic disease, respectively. Subsequently, a series of functional experiments were conducted to examine the effect of the transfection on the cells grown in adherent conditions and colonospheres of both cell lines. First, we evaluated the characteristics of CSCs obtained by sphere culture, wherein CSCs are trapped and enriched, which is an extremely effective CSC isolation method for cancer cell lines [26]. The expression levels of miR-486-5p in adherent cells and colonospheres were evaluated by qPCR; data confirmed its downregulation in spheroid culture with respect to adherent culture condition in both cell lines [25]. Afterwards, we assessed the presence of CSCs in spheroid culture models evaluating the ALDH1 activity, a validated CSC marker in various tumors, including CRC [38,39,40]. ALDH1-positive cell percentages were observed to differ between the two cell lines grown in adherence, as reported also by Alowaidi et al., who identified differences in ALDH1 percentage between two colon cell lines [41]. When we compared ALDH1 activity between the two culture models, our results showed a higher percentage of ALDH1-positive cells in spheroid culture models of both cell lines with respect to adherence cultured cells. This indicates that both lines were capable of forming colonospheres. After treatments with a miR-486-5p mimic and inhibitor, it was observed that ALDH1 activity was affected by transfection in a different manner in the two cell lines. The results of the T84 line experiments confirm the tumor-suppressor role of miR-486-5p, showing its impact not only on tumor growth but also on the phenotype of CSCs. In HT-29 cells, we observed an increase in ALDH1 in treated cells compared to the control, both in adherent cells and colonospheres. However, in adherent cells, a slight increase of ALDH1-positive cells was observed when treated with the inhibitor rather than with the mimic; the same was not observed in the colonospheres, where almost identical levels of ALDH1 were observed between cells treated with the inhibitor and those treated with the mimic. The increase in ALDH1 in treated cells with respect to the control may be related to a response to treatment; in the literature, it has been observed that in some tumors, its levels can increase in response to treatments [42,43]. CSCs are also characterized by high levels of stemness- and EMT-related genes [44,45]. To characterize the effect of miR-486-5p on stemness and EMT, we compared the expression levels of stemness genes, including *OCT-4*, *SOX-2*, *c-Myc*, and *KLF-4*, and EMT genes, including *Vimentin*, *SNAIL*, and *SLUG*, in both spheroids and parental cells. MiR-486-5p influences both EMT and stemness in cancer by targeting key regulators. It inhibits EMT by suppressing genes like ZEB1 and *Vimentin* while promoting E-cadherin expression. For stemness, miR-486-5p can modulate transcription factors such as *OCT-4* and *SOX-2*, reducing self-renewal and tumor-initiating capabilities. These interconnected effects highlight its role as a tumor suppressor, though its impact may vary across different contexts. Further studies are essential to clarify these mechanisms [36,46,47]. The transcription factors *SOX-2*, *OCT-4*, *c-Myc*, and *KLF-4* are known to play critical roles in the regulation of gene expression and are particularly important in the context of cell reprogramming and induced pluripotent stem cells (iPSCs). They have been shown to be overexpressed in CSCs of various tumor types and are therefore used as CSC biomarkers [26,48,49,50,51,52]. When we assessed the expression levels of CSC biomarkers in colonospheres obtained from HT-29 and T84 cell lines, *SOX-2* and *OCT-4* resulted to be overexpressed in HT-29 colonospheres respect to adherent cells, and *OCT-4*, *SOX-2*, *c-Myc*, and *KLF-4* resulted to be overregulated in T84 colonospheres in comparison with the adherent counterpart. These data were a further confirmation of the presence of CSCs in our spheroid culture model. To evaluate the effects of miR-486-5p transfection in our culture models (monolayer and spheres culture models), we assessed the same transcription factors in mimic-, inhibitor-transfected, and nontransfected cells. In our study, the effect of miR-486-5p on *OCT-4* as a stemness inhibitor presents complex and context-dependent results. Specifically, in HT-29 monolayers, cells treated with the mimic displayed lower *OCT-4* expression compared to those treated with the inhibitor. Conversely, in HT-29 colonospheres, and in both conditions of T84 cells, the results were opposite, with *OCT-4* levels increasing following miR-486-5p transfection in HT-29 colonospheres. This phenomenon may reflect an adaptive resistance mechanism, as suggested by previous reports where *OCT-4* levels increased in response to combined treatments with BEZ235 and miR-21 inhibitors [53]. Additionally, miR-486-5p modulated *SOX-2* expression differently depending on the context. In HT-29 adherent cells and colonospheres, miR-486-5p increased *SOX-2* levels, though significance in colonospheres was observed only when compared to the inhibitor condition. By contrast, in T84 adherent cells, *SOX-2* expression was reduced by miR-486-5p, while inhibition of the miRNA led to higher *SOX-2* levels. This suggests that miR-486-5p might exert cell-type-specific effects on *SOX-2*, which could play varying roles in tumorigenesis. For instance, prior research has shown that silencing *SOX-2* promotes larger tumor development, whereas its overexpression correlates with smaller tumors, a finding consistent with our clonogenic assays showing smaller colonies at higher *SOX-2* expression levels [54,55]. These observations underline the complexity of miR-486-5p’s interaction with stemness markers such as OCT-4 and SOX-2, highlighting context-specific regulatory mechanisms. However, the exact pathways underlying these effects remain unclear, and further studies are necessary to validate and expand upon these findings, with a focus on the molecular mechanisms and their implications for therapeutic applications. The expression levels of *c-Myc* were found to be higher in both colonospheres models when miR-486-5p was inhibited, in comparison with colonospheres in which miRNA overexpression was simulated. This effect appears to support the hypothesis that miR486-5p plays a protective role with respect to the CSC phenotype [56,57,58]. This finding is consistent with what we observed in our previous work and with the literature, as other authors have highlighted its involvement in CSCs [25,58] as well as the influence of miR-486-5p on *c-Myc* in CRC [59]. *Vimentin*, *SNAIL*, and *SLUG* overexpression was demonstrated in several tumors, including CRC [60,61,62,63,64,65,66]. Due to their role in epithelial-to-mesenchymal transition (EMT), miRNAs are considered markers of this process [67,68,69]. When assessing the expression levels of EMT biomarkers in colonospheres obtained from HT-29 and T84 cell lines, the data showed that *Vimentin* was overexpressed in HT-29 colonospheres compared to the adherent counterpart, while SLUG and Vimentin were overexpressed in T84 colonospheres compared to T84 adherent cells. *Vimentin* is not only involved in EMT but also associated with CSCs, promoting resistance to apoptosis and tumor invasion, key elements in metastatic progression. Moreover, *Vimentin* regulates CSC responses to fractionated radiation exposure in cervical cancer, highlighting its role in the survival and proliferation of these stem cells [70]. In colorectal cancer, *Vimentin*-positive circulating tumor cells have been shown to be a prognostic biomarker for the advanced stages of the disease [71].

Recent studies have shown that tumor cells gain mesenchymal traits sequentially while retaining some of the previously expressed epithelial characteristics [72]. Furthermore, Jolly, Jia, and colleagues demonstrated that partial EMT is linked to stemness [73]. How miR-486-5p affected the EMT process is not clear. The distinctive mutations of our models (such as BRAF and PIK3CA) directly regulate the expression levels of EMT markers via the same pathway that appears to be affected by the inhibition or overexpression of miR-486-5p [74,75,76,77,78]. *Vimentin* expression resulted in being significantly downregulated in HT-29 colonospheres but without statistical difference between mimic and inhibitor miR-486-5p transfection, and no statistical differences resulted after the transfections in monolayer models and T84 colonospheres. miRNA-486-5p probably was not directly related to *Vimentin* expression levels, but it can influence *Vimentin* expression in this specific model (HT-29 colonospheres). The inhibition of miR-486-5p promoted *SNAIL* expression in HT-29 adherent cells, while in T84 adherent cell models it was seen to suppress *SNAIL* expression. *SLUG* resulted in being downregulated in all culture models after transfections with both miR-486-5p mimic and inhibitor, but the miRNA exerts its suppressive role in the T84 colonosphere model in which miR-486-5p inhibition increases *SLUG* expression compared to colonosphere in which we simulated the miRNA overexpression. These findings suggest the existence of differences in the involved and activated pathways between the two cell lines, which are plausibly caused by the presence of different genetic mutations [79]. The results collectively indicate that miR-486-5p exerts a complex and variable influence on the pathways regulating stemness- and EMT-associated transcription factors in our in vitro models. Its effect may vary not only between different cell lines but also between different culture conditions, suggesting that cellular context and microenvironment play a critical role in determining the efficacy of miR-486-5p as a tumor suppressor. Another necessary consideration concerns the fact that the maintenance of the CSC phenotype, despite the absence of upregulation of all stemness and EMT markers, can be attributed to the flexibility of stemness networks. Indeed, evidence suggests that certain transcription factors can induce a stem phenotype even when acting individually, through the activation of specific sets of human pluripotency regulators [80]. In our study, we conducted the expression analysis of multiple stemness markers but probably not the combination of markers responsible for maintaining the stemness characteristics observed in transfected colonospheres generated in our laboratory. Furthermore, it is essential to consider that miR-486-5p manipulations may lead to potential off-target effects, such as unintended regulation of non-target mRNAs due to partial sequence complementarity. These interactions could influence cellular pathways unrelated to the study’s primary focus. Overexpression or inhibition of miR-486-5p may also disrupt the broader miRNA network, potentially triggering widespread cellular responses. To reduce potential off-target effects, we used appropriate controls, including scramble miRNA and mock transfections. These controls helped isolate the specific effects of miR-486-5p manipulations, ensuring that observed results were not due to nonspecific interactions or the transfection procedure itself.

The impact of transfection was most pronounced when the colonospheres were cultivated in soft agar to evaluate their colony-forming capacity, which is an indicator of cell metastatic potential [81,82]. This technique allowed us to examine the ability of a single cell to grow into a large colony through clonal expansion and to expand CSCs [81,83]. The results of our experiments confirmed that miR-486-5p exerts a suppressive effect on HT-29 colonies. This was highlighted by the fact that when the miRNA content was increased using the mimic, the size of the colonies decreased, contrary to what happened with the use of the inhibitor. Additionally, we observed a correlation between changes in colony size and alterations in the number of colonies. Specifically, in our results, a decrease in colony size corresponded to an increase in the number of colonies and vice versa. The increase in the number of colonies when miR-486-5p is overexpressed would seem to indicate an oncogenic power of this miRNA and its role as a promoter of stemness. All in all, this data is in contrast with the other phenomena observed and with what happens in T84; therefore, we interpreted it as the result of competition for nutrients: the number of colonies present may be attributed to the equilibrium between the available nutrients and the replication rate of the cells. miRNA 486-5p does not appear to have exerted a direct influence on the viability of the cells, as evidenced by the outcomes of the Alamar blue cell viability assay [84]. However, we postulate that miRNA was capable of limiting the proliferation of cells that, having survived the miR-486-5p treatment, still formed colonies but proliferated at a low rate, resulting in small colonies. On the other hand, miRNA inhibition may have enhanced the proliferative capacity of cells that were able to proliferate by forming large colonies at the expense of other cells that, due to nutrient scarcity, did not form colonies. So, there is an increase in proliferation, but this is limited by nutrient availability, which gives an advantage to the larger colonies. This hypothesis would elucidate why the viability assay demonstrated no difference in the viability rate of colonospheres despite the evident increase in colony size when miRNA was inhibited and the reduction in size when miRNA overexpression was simulated. This interpretation aligns with previous studies that have highlighted the role of miRNAs in modulating stem cell properties and tumor growth. For instance, Chakraborty et al. emphasized that miRNAs can influence the balance between proliferation and differentiation in stem cells, affecting their tumorigenic potential [85]. Moreover, studies by Lobel et al. [84] demonstrated that nutrient availability plays a crucial role in regulating cell proliferation and colony formation in cancer models. The effect of transfection in T84 colonies was less pronounced yet exhibited a comparable trend. Once more, a reduction in colony size was observed following the simulated overexpression of miR-486-5p, though not to the same extent as in HT-29. It may be hypothesized that this discrepancy can be attributed to the fact that, as this was an already metastatic line, more active metastatic signals were present, which attenuated the suppressive power of miR-486-5p. Furthermore, in this case, there appears to be no competition for nutrients, and a reduction in colony size was accompanied by a decline in colony number. Once more, this could be attributed to the relatively limited impact of miRNA on this particular cell line, given the diverse metastatic active signals. Consequently, the effect of miRNA was insufficient to markedly reduce or increase the viability rate of these cells, although a slight reduction in colony size was observed. Considering that clonogenic activity represents a highly sensitive indicator of undifferentiated CSCs and that colony size is directly correlated with cell proliferation, our results suggest that miR-486-5p has the capacity to inhibit CSC proliferation, resulting in reduced colony size [81,86].

## 5. Conclusions

The present study demonstrates that miR-486-5p plays a role in CCSC proliferation and, consequently, in the promotion of CRC. Our in vitro experiments suggest that miR-486-5p affects pathways regulating stemness and EMT transcription factors, with varying outcomes between cell lines and culture conditions. This highlights the critical role of the cellular context and microenvironment in miR-486-5p efficacy as a tumor suppressor. Based on our previous studies and on the results obtained in this work with the colony formation assay, in which we obtained an increase in colony size with the inhibition of miR-486-5p and a decrease when we simulated its overexpression, we can assume that miR-486-5p does effectively affect the stemness phenotype. These results encourage us to carry out in vivo experiments to gain further insight into the real role of miR-486-5p on progression and metastasis in CRC, which could have a future clinical impact in both the diagnosis and treatment.

## Figures and Tables

**Figure 1 cancers-16-04237-f001:**
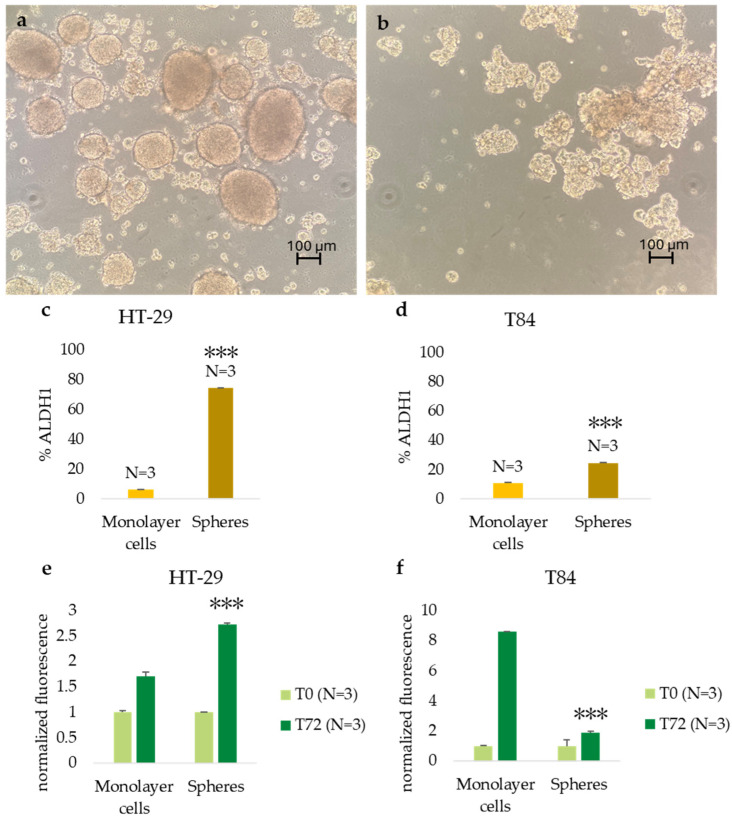
Representative image of secondary spheres formed from HT29 cell lines (**a**), and from T84 cell lines (**b**) obtained in serum-free, nonadherent conditions. Image captured with 10× lens; ALDH1-positive cell percentage in adherence-grown cells and spheres obtained from HT-29 cell lines (**c**), and from T84 cell lines (**d**); cell viability rate of adherence-grown cells and spheres obtained from HT-29 cell lines (**e**), and from T84 cell lines (**f**). Representative flow cytometry plot of ALDH1 activity in monolayer cells and spheres; *x*-axis: FITC (ALDH1), *y*-axis: SSC-A (**g**). The symbol *** indicates the statistical difference between monolayer cells and spheres with *p*-value < 0.001. Values expressed as Mean ± SE. All analyses were performed in triplicate. N = sample size for each analysis.

**Figure 2 cancers-16-04237-f002:**
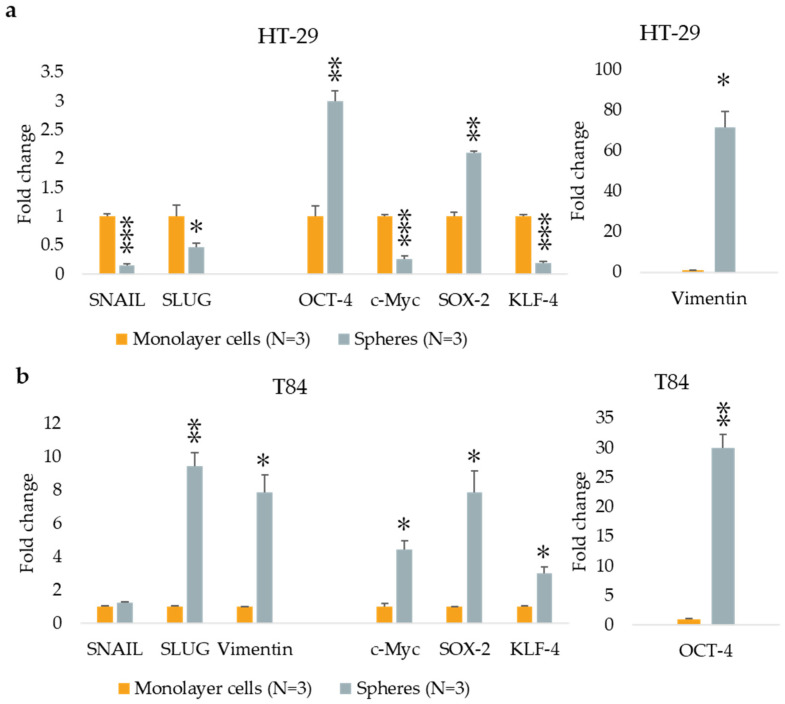
EMT and stemness marker expression in monolayer cells and in spheres obtained from HT-29 cell lines (**a**) and from T84 cell lines (**b**). The symbol * indicates the statistical difference between monolayer cells and spheres with a *p*-value < 0.05. The symbol ** indicates the statistical difference between monolayer cells and spheres with a *p*-value < 0.01. The symbol *** indicates a statistical difference between monolayer cells and spheres with a *p*-value < 0.001. Values expressed as Mean ± SE. All analyses were performed in triplicate. N = sample size for each analysis.

**Figure 3 cancers-16-04237-f003:**
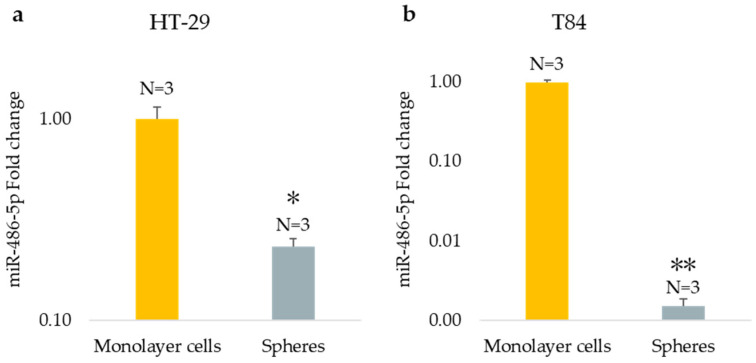
miRNA-486-5p expression in monolayer cells and in spheres obtained from HT-29 cell lines (**a**) and from T84 cell lines (**b**). The symbol * indicates the statistical difference between monolayer cells and spheres with a *p*-value < 0.05. The symbol ** indicates the statistical difference between monolayer cells and spheres with a *p*-value < 0.01. Values expressed as Mean ± SE. All analyses were performed in triplicate. N = sample size for each analysis.

**Figure 4 cancers-16-04237-f004:**
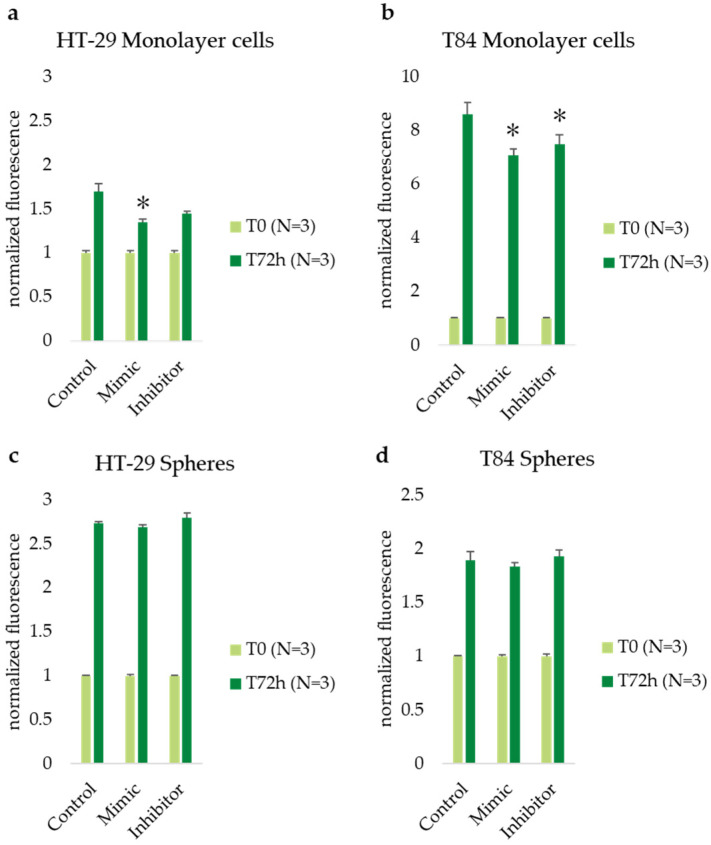
Cell viability assessed at time 0 (T0) and after 72 h (T72) of miR-486-5p transfection in HT-29 and T84 monolayer (**a**,**b**); cell viability at time 0 (T0) and after 72 h (T72) of miR-486-5p transfection in spheres obtained from HT-29 and T84 cell lines (**c**,**d**). The symbol * indicates the statistical difference between control and treated cells with a *p*-value < 0.05. Values expressed as Mean ± SE. All analyses were performed in triplicate. N = sample size for each analysis.

**Figure 5 cancers-16-04237-f005:**
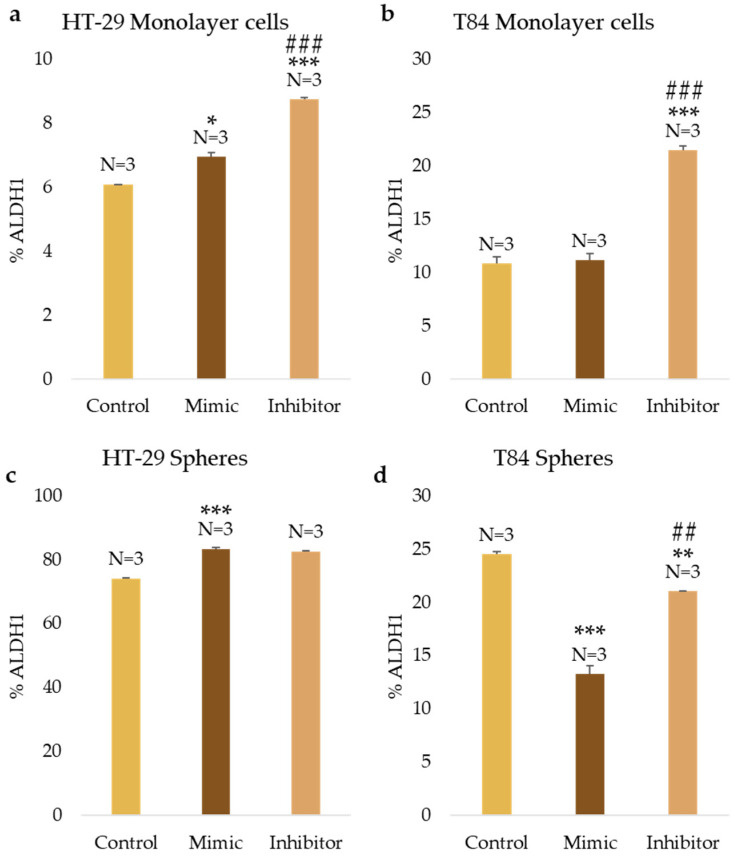
ALDH1 positive cell percentage after the transfection of miR-486-5p mimic and inhibitor in HT-29 and T84 monolayer (**a**,**b**); ALDH1 positive cell percentage after the transfection of miR-486-5p mimic and inhibitor in HT-29 and T84 spheres (**c**,**d**). Representative flow cytometry plot of ALDH1 activity in monolayer cells and spheres after the transfection of miR-486-5p mimic and inhibitor; *x*-axis: FITC (ALDH1), *y*-axis: SSC-A (**e**). The symbol * indicates the comparison between control and treated cells with a *p*-value < 0.05. The symbol ** indicates the comparison between control and treated cells with a *p*-value < 0.01. The symbol *** indicates the comparison between control and treated cells with a *p*-value < 0.001. The symbol ## indicates the statistical difference between mimic- and inhibitor-treated cells with a *p*-value < 0.01. The symbol ### indicates the statistical difference between mimic- and inhibitor-treated cells with a *p*-value < 0.001. Values expressed as Mean ± SE. All analyses were performed in triplicate. N = sample size for each analysis.

**Figure 6 cancers-16-04237-f006:**
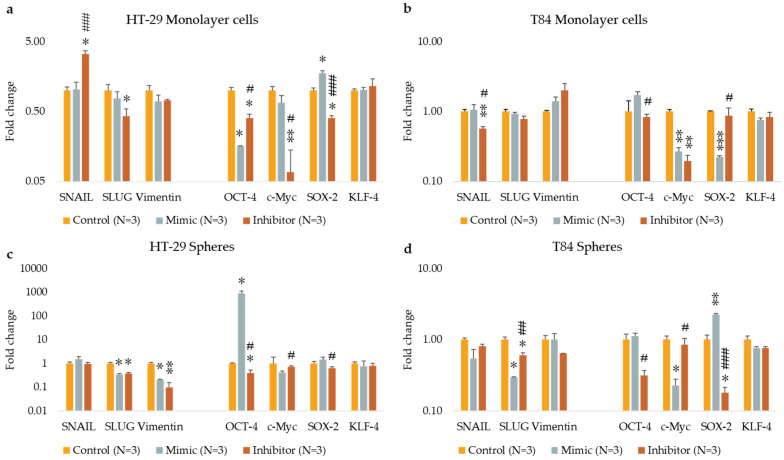
EMT- and stemness-related gene expression after the transfection of miR-486-5p mimic and inhibitor in HT-29 and T84 monolayer (**a**,**b**); EMT- and stemness-related gene expression after the transfection of miR-486-5p mimic and inhibitor in HT-29 and T84 spheres (**c**,**d**). The symbol * indicates the comparison between control and treated cells with a *p*-value < 0.05. The symbol ** indicates the comparison between control and treated cells with a *p*-value < 0.01. The symbol *** indicates the comparison between control and treated cells with a *p*-value < 0.001. The symbol # indicates the statistical difference between mimic- and inhibitor-treated cells with a *p*-value < 0.05. The symbol ## indicates the statistical difference between mimic- and inhibitor-treated cells with a *p*-value < 0.01. The symbol ### indicates the statistical difference between mimic- and inhibitor-treated cells with a *p*-value < 0.001. Values expressed as Mean ± SE. All analyses were performed in triplicate. N = sample size for each analysis.

**Figure 7 cancers-16-04237-f007:**
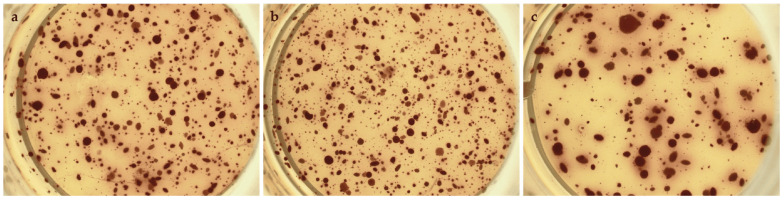
Representative images of HT-29 untreated cells (**a**), mimic-treated cells (**b**), and inhibitor-treated cells (**c**) in soft agar colony formation assay. Comparison of the number of colonies of untreated, mimic-treated, and inhibitor-treated HT-29 cells (**d**). Comparison of average colony area of untreated, mimic-treated, and inhibitor-treated HT-29 cells (**e**). Percentage of colonies divided by size (**f**). A comparison of the area and number of colonies for HT-29 spheres (**g**). The symbol * indicates the comparison between control and treated cells with a *p*-value < 0.05. The symbol ** indicates the comparison between control and treated cells with a *p*-value < 0.01. The symbol *** indicates the comparison between control and treated cells with a *p*-value < 0.001. The symbol ## indicates the statistical difference between mimic- and inhibitor-treated cells with a *p*-value < 0.01. The symbol ### indicates the statistical difference between mimic- and inhibitor-treated cells with a *p*-value < 0.001. Values expressed as Mean ± SE. All analyses were performed in triplicate. N = sample size for each analysis.

**Figure 8 cancers-16-04237-f008:**
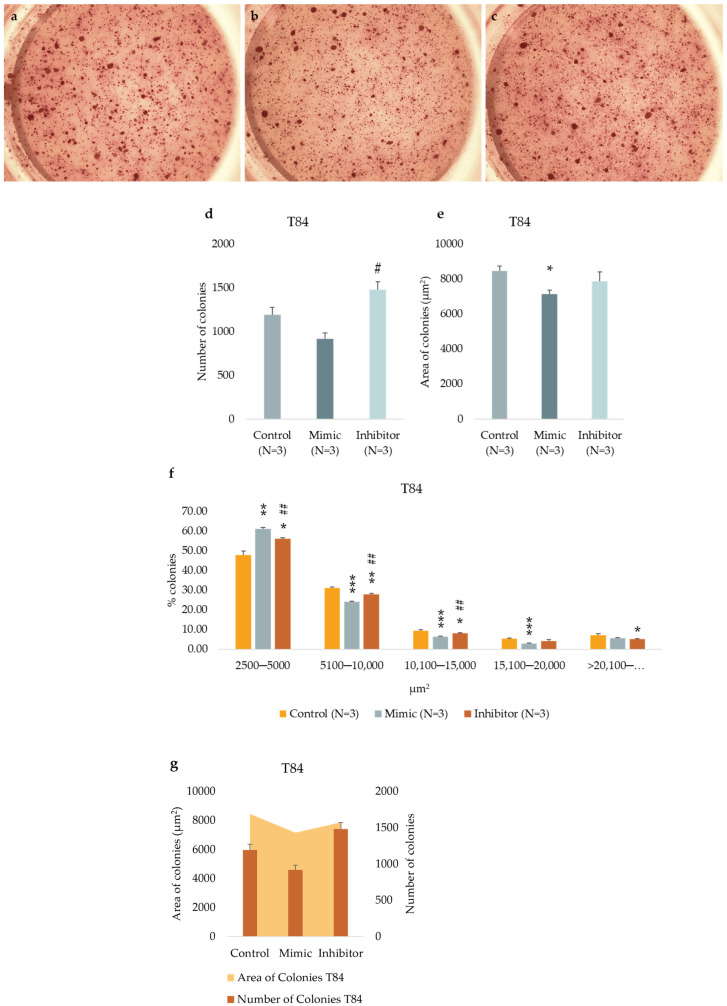
Representative images of T84 untreated cells (**a**), mimic-treated cells (**b**), and inhibitor-treated cells (**c**) in soft agar colony formation assay. Comparison of the number of colonies of untreated, mimic-treated and inhibitor-treated T84 cells (**d**). Comparison of average colony area of untreated, mimic-treated and inhibitor-treated T84 cells (**e**). Percentage of colonies divided by size (**f**). A comparison of the area and number of colonies for T84 spheres (**g**). The symbol * indicates the comparison between control and treated cells with a *p*-value < 0.05. The symbol ** indicates the comparison between control and treated cells with a *p*-value < 0.01. The symbol *** indicates the comparison between control and treated cells with a *p*-value < 0.001. The symbol # indicates the statistical difference between mimic- and inhibitor-treated cells with a *p*-value < 0.05. The symbol ## indicates the statistical difference between mimic- and inhibitor-treated cells with a *p*-value < 0.01. Values expressed as Mean ± SE. All analyses were performed in triplicate. N = sample size for each analysis.

## Data Availability

Data are contained within the article.

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
