# Peer review of "The Role of miR-486-5p on CSCs Phenotypes in Colorectal Cancer"

_cancers, 2024, doi:10.3390/cancers16244237_

Round 1
Reviewer 1 Report
Comments and Suggestions for Authors
Recommendation: Major review
In this manuscript, the author reports, " Role of miR-486-5p on CSCs phenotype in colorectal cancer ". The authors should address the following questions before getting publication.
1. provide more details on how the two CRC cell lines were selected and their relevance to studying miR-486-5p in CSCs?
2. clarify the criteria used for choosing the specific miRNAs for investigation in this study?
3. Is there a particular reason for using only two CRC cell lines? Would additional cell lines help strengthen the findings?
4. How was the concentration of miR-486-5p mimic and inhibitor determined for the transfection experiments?
5. elaborate on the validation process of the transfection efficiency for both miR-486-5p mimic and inhibitor?
6. In the section on cell viability assays, how were the reference controls selected, and how do they validate the results?
7. The paper discusses the overexpression and underexpression of miR-486-5p. Could you explain how these different expressions impact CSCs specifically in colorectal cancer?
8. Discuss any potential off-target effects of the miR-486-5p manipulations?
9. In the discussion, the authors mention "stemness" and "EMT genes." Could they provide a more detailed mechanistic explanation of how these genes interact with miR-486-5p?
Comments on the Quality of English LanguageRecommendation: Major review
In this manuscript, the author reports, " Role of miR-486-5p on CSCs phenotype in colorectal cancer ". The authors should address the following questions before getting publication.
1. provide more details on how the two CRC cell lines were selected and their relevance to studying miR-486-5p in CSCs?
2. clarify the criteria used for choosing the specific miRNAs for investigation in this study?
3. Is there a particular reason for using only two CRC cell lines? Would additional cell lines help strengthen the findings?
4. How was the concentration of miR-486-5p mimic and inhibitor determined for the transfection experiments?
5. elaborate on the validation process of the transfection efficiency for both miR-486-5p mimic and inhibitor?
6. In the section on cell viability assays, how were the reference controls selected, and how do they validate the results?
7. The paper discusses the overexpression and underexpression of miR-486-5p. Could you explain how these different expressions impact CSCs specifically in colorectal cancer?
8. Discuss any potential off-target effects of the miR-486-5p manipulations?
9. In the discussion, the authors mention "stemness" and "EMT genes." Could they provide a more detailed mechanistic explanation of how these genes interact with miR-486-5p?
Author Response
Comment 1: provide more details on how the two CRC cell lines were selected and their relevance to studying miR-486-5p in CSCs?
Answer 1:
Authors: Thank you for your observation. We have added in material and methods more details about our cell lines used.
Line 151-160
We chose these two cell lines because they represent distinct stages of colorectal cancer progression. HT29 models a well-differentiated primary colorectal adenocarcinoma with high proliferative capacity and morphology closely resembling the original tumor. In contrast, T84, derived from liver metastases, serves as a representative model of metastatic colorectal cancer, enabling the study of tumor progression mechanisms. These lines may exhibit differences in microRNA expression patterns, allowing comparisons between primary and metastatic tumors. For example, microRNAs associated with metastasis or chemoresistance might be more expressed in T84. The use of these complementary models allows for the investigation of diagnostic or therapeutic markers in a context that mirrors the clinical progression of the disease.
Furthermore, both cell lines are widely utilized in colorectal cancer research, supported by extensive literature that ensures comparability with prior studies (https://doi.org/10.1158/0008-5472.CAN-17-1869; https://journals.plos.org/plosone/article?id=10.1371/journal.pone.0242057)
Comment2: Clarify the criteria used for choosing the specific miRNAs for investigation in this study?
Answer2:
Authors: Thank you for your observation. We have emphasized and clarified in the introduction the use of this miRNA for our study, which, as you can see, has been the subject of previous studies in our colorectal cancer research.
Line 130-137
In our previous research, we identified miR-486-5p as a promising candidate for CRC biomarker studies based on its differential expression in CRC patient samples (serum and stool) across tumor stages, along with our meta-analysis using GEO datasets. Our findings showed that miR-486-5p was downregulated in metastatic patients and in CSC models, compared to healthy controls and parental cells. This miRNA’s stability in biological fluids, coupled with its deregulation in CRC, makes it a promising candidate for diagnostic and prognostic applications in CRC, motivating our continued investigation [25].
Comment 3: Is there a particular reason for using only two CRC cell lines? Would additional cell lines help strengthen the findings?
Answer 3:
Authors: We appreciate your consideration and we would like to explain that the lines used have already been used in the previous study and their choice is linked to the fact that they are representative of two conditions of the pathology, the HT29 of the primary tumor and the T84 of the metastatic tumor. The present work aims to better clarify the aspects highlighted in the previous work.
Like other author, we used just two cell lines: DOI: 10.1186/s11658-020-00243-8; https://link.springer.com/article/10.1023/A:1005986606010; doi: 10.3389/fimmu.2022.947136; doi: 10.1186/s12951-019-0563-2
Comment 4: How was the concentration of miR-486-5p mimic and inhibitor determined for the transfection experiments?
Answer 4:
Authors: Thank you for the question, we describe you the methodology that we follow to our experiments:
In order to obtain the best concentration for transfection we tested a range of concentrations using mimic and inhibitor associated with a fluorescent probe. For mimics, we evaluated 0.015 µL (2.5 nM), 0.03 µL (5 nM), 0.06 µL (10 nM), 0.12 µL (20 nM), 0.18 µL (30 nM), and 0.24 µL (40 nM). For inhibitors, we tested 0.25 µL (25 nM), 0.5 µL (50 nM), 0.6 µL (60 nM), 0.75 µL (75 nM), and 1 µL (100 nM).
In accordance with the indication of the manufacturing instructions, we observed the cells daily over five days, assessing fluorescence levels and signs of cytotoxicity. At the end of the experiment, we counted viable cells to evaluate toxicity at higher reagent concentrations. This analysis allowed us to balance fluorescence recovery and cell viability, identifying the optimal concentration range for both mimics and inhibitors. The best results were obtained with final concentrations of 5 nM for mimics and 50 nM for inhibitors, achieving robust fluorescence signals with minimal cytotoxicity.
Furthermore, we tested the concentration of miR-486-5p before and after transfections by real time PCR, to confirm the effective transfection.
To clarify this point, we have referenced in material and methods our previous work.
Comment 5: elaborate on the validation process of the transfection efficiency for both miR-486-5p mimic and inhibitor?
Answer 5:
Authors: We gave the answer in the previous question, see answer 4.
Comment 6: In the section on cell viability assays, how were the reference controls selected, and how do they validate the results?
Answer 6:
Authors: In that case the control was represented by the untreated cells.
Comment 7: The paper discusses the overexpression and underexpression of miR-486-5p. Could you explain how these different expressions impact CSCs specifically in colorectal cancer?
Answer 7:
Authors: Thank you for the observation we have included at the discussion.
Initially, we observed a downregulation of miR-486-5p across all CSC models [lines 593, 595], followed by its effect on ALDH1 expression [lines 595-602]. The impact of miR-486-5p on the CSC phenotype is further discussed in relation to stemness factors and EMT, highlighting its complex, context-dependent nature. In some cases, miR-486-5p reduces stemness markers like c-Myc, suggesting a protective role. However, in others, such as HT-29 colonospheres, it upregulates markers like OCT-4, possibly as an adaptive response. This underscores the influence of the microenvironment and cellular context [lines 617-705]. Finally, soft agar assays revealed changes in the colonosphere morphology following miR-486-5p transfection or inhibition, providing additional evidence of its effect on the CSC phenotype. Our previous work also demonstrated miR-486-5p's involvement in ALDH1, stemness genes, and the regulation of key stemness pathways, including Wnt, Notch, Hedgehog, and TGF-β [line 565-570].
Comment 8: Discuss any potential off-target effects of the miR-486-5p manipulations?
Answer 8:
Authors: Thank you for the observation we improved the discussion.
Line 697-705:
Furthermore, it is essential to consider that miR-486-5p manipulations may lead to potential off-target effects, such as unintended regulation of non-target mRNAs due to partial sequence complementarity. These interactions could influence cellular pathways unrelated to the study's primary focus. Overexpression or inhibition of miR-486-5p may also disrupt the broader miRNA network, potentially triggering widespread cellular responses. To reduce potential off-target effects, we used appropriate controls, including scramble miRNA and mock transfections. These controls helped isolate the specific effects of miR-486-5p manipulations, ensuring that observed results were not due to nonspecific interactions or the transfection procedure itself.
Comment 9: In the discussion, the authors mention "stemness" and "EMT genes." Could they provide a more detailed mechanistic explanation of how these genes interact with miR-486-5p?
Answer 9:
Authors: Thank you for your observation, we modified the discussion adding some references about the mechanistic effect of miR-486-5 on EMT and stemness genes:
Line 606-612: miR-486-5p influences both EMT and stemness in cancer by targeting key regulators. It inhibits EMT by suppressing genes like ZEB1 and vimentin while promoting E-cadherin expression. For stemness, miR-486-5p can modulate transcription factors such as OCT-4 and SOX-2, reducing self-renewal and tumor-initiating capabilities. These interconnected effects highlight its role as a tumor suppressor, though its impact may vary across different contexts. Further studies are essential to clarify these mechanisms [38,48,49] (doi: 10.1186/s12885-024-13086-9, doi: 10.1158/0008-5472.CAN-19-1624, doi: 10.1186/s12957-021-02381-5).
Reviewer 2 Report
Comments and Suggestions for Authors
Etzi et al reported a study on the role of miR-486-5p on the stemness of colorectal cancer cells, and suggested a tumor-suppressive function. miR-486-5p was down-regulated in stem cell enriched spheres and over expression of miR-486-5p reduced stem cell features in T84 spheres. However, there are several issues to address before it comes to publication.
Major points:
1. All the flow cytometry charts are not properly plotted. The Alamar Blue assay requires measurement of fluorescence intensity, rather than FSC/SSC in current figures. Authors only gated cells by size/complexity but didn’t show any fluorescence information. A proper chart would be, for example, a histogram of the fluorescence intensity on gated cells. Besides, authors may also include a live/dead dye to gated on live cells.
2. All the cell viability figures are confusion as the readings exceed 100%. Since authors cultured the cells for 72 hours, those results are more likely cell proliferation instead of viability. To evaluate cell viability authors should count all cells and live cells, and then calculate the live/total ratio.
3. All the gene expression figures are already on a log-scale y axis. Therefore, the y-axis label should be simply “fold change”.
4. Overexpression of miR-486-5p appears to have opposite effect on the enrichment of stem cells in spheres (figure 5). It decreased stem cell ratio (%ALDH1) in T84 spheres but increased %ALDH1 in HT-29 spheres. Overexpression of miR-486-5p also promoted colony formation in HT-29 cells (figure 7). Authors should discuss such inconsistency in the results, otherwise the conclusion drawn from current experiments would be that miR486-5p is sometimes onco-suppressive and sometimes oncogenic, which is context-dependent. Authors may need to test more cell lines or use some in vivo animal models. The inconsistency in stem cell markers OCT-4 and SOX-2 should also be discussed.
Minor point:
Authors should include a protocol for the flow cytometry and analysis strategies.
Author Response
Comment 1. All the flow cytometry charts are not properly plotted. The Alamar Blue assay requires measurement of fluorescence intensity, rather than FSC/SSC in current figures. Authors only gated cells by size/complexity but didn’t show any fluorescence information. A proper chart would be, for example, a histogram of the fluorescence intensity on gated cells. Besides, authors may also include a live/dead dye to gated on live cells.
Answer 1.
Authors: Thanks to the reviewer for the observation.
We have reviewed and corrected the ALDH1 plots as they were previously misrepresented. Regarding the Alamar Blue assay, we want to clarify that we measured cell viability as described in the Materials and Methods section, using a spectrophotometer to assess the fluorescence intensity emitted by cells treated with the reagent.
Comment 2. All the cell viability figures are confusion as the readings exceed 100%. Since authors cultured the cells for 72 hours, those results are more likely cell proliferation instead of viability. To evaluate cell viability authors should count all cells and live cells, and then calculate the live/total ratio.
Answer 2.
Authors: We appreciate the reviewer’s observation and have revised our analysis to ensure clarity regarding cell viability. To address this, we have corrected the representation of our data for improved clarity and labeled it explicitly as metabolic activity indicative of cell viability.
The Alamar Blue assay is based on the reduction of resazurin to resorufin within viable cells, a process that is dependent on metabolic activity. This reaction generates fluorescence, which is directly proportional to the number of viable cells. Thus, the assay is widely used to assess cell viability because it provides an accurate measure of metabolic activity, is sensitive, and can be performed quickly and non-invasively, making it an ideal tool for our viability measurements. Therefore, we used the Alamar Blue assay to assess cell viability, as it provides a reliable estimate of the number of live cells present in the sample, which aligns with the goals of our study.
Comment 3. All the gene expression figures are already on a log-scale y axis. Therefore, the y-axis label should be simply “fold change”.
Answer 3.
Authors: Thanks to the reviewer for the appreciation. We have modified the data and the graphics, and have included fold change. To make the data clearer in Figure 2, we divided the graph and represented onlyVimentin in HT-29 and OCT-4 in the T84 cell lines.
Comment 4. Overexpression of miR-486-5p appears to have the opposite effect on the enrichment of stem cells in spheres (figure 5). It decreased stem cell ratio (%ALDH1) in T84 spheres but increased %ALDH1 in HT-29 spheres. Overexpression of miR-486-5p also promoted colony formation in HT-29 cells (figure 7). Authors should discuss such inconsistency in the results, otherwise the conclusion drawn from current experiments would be that miR486-5p is sometimes onco-suppressive and sometimes oncogenic, which is context-dependent. Authors may need to test more cell lines or use some in vivo animal models.
The inconsistency in stem cell markers OCT-4 and SOX-2 should also be discussed.
Answer 4.
Authors: We appreciate the commentary of the reviewer,
About the figure 5: we can say that in HT-29 cells the augment of ALDH1 may be attributed to the treatment, in fact we observe an augment of ALDH1 in both mimic and inhibitor treated cells (also if in the last case it is not significative, but a also we didn’t find significative difference between mimic and inhibitor). So this data alone does not offer a specific role of miR-486-5p on this line.
About the figure 7: Our results indicate that miR-486-5p overexpression in HT-29 cells increased the number of colonies formed, as shown in Figure 7. However, this was accompanied by a significant reduction in colony size, suggesting a potential trade-off between the number and size of colonies, likely driven by competition for limited nutrients. This dynamic may reflect miR-486-5p's capacity to modulate proliferation, limiting the growth rate of individual cells while promoting the formation of smaller but more numerous colonies. Regarding T84 cells, we observed a less pronounced effect on colony number and size compared to HT-29 cells. This difference might be attributed to the distinct genetic and metastatic profiles of T84 cells, which are already highly metastatic. Their active signaling pathways likely attenuate the suppressive effects of miR-486-5p. Moreover, the lack of evident nutrient competition in T84 cultures may explain why reductions in colony size were not accompanied by increased colony numbers, unlike in HT-29 cells. This underscores the context-dependent effects of miR-486-5p, influenced by cellular model and microenvironment. We improved the text [line 714-723]
Regarding the results on SOX-2 and OCT-4, we have considered your feedback and have rewritten the discussion.
Line 624-646: In our study, the effect of miR-486-5p on OCT-4 as a stemness inhibitor presents complex and context-dependent results. Specifically, in HT-29 monolayers, cells treated with the mimic displayed lower OCT-4 expression compared to those treated with the inhibitor. Conversely, in HT-29 colonospheres and in both conditions of T84 cells, the results were opposite, with OCT-4 levels increasing following miR-486-5p transfection in HT-29 colonospheres. This phenomenon may reflect an adaptive resistance mechanism, as suggested by previous reports where OCT-4 levels increased in response to combined treatments with BEZ235 and miR-21 inhibitors [55]. Additionally, miR-486-5p modulated SOX-2 expression differently depending on the context. In HT-29 adherent cells and colonospheres, miR-486-5p increased SOX-2 levels, though significance in colonospheres was observed only when compared to the inhibitor condition. By contrast, in T84 adherent cells, SOX-2 expression was reduced by miR-486-5p, while inhibition of the miRNA led to higher SOX-2 levels. This suggests that miR-486-5p might exert cell-type-specific effects on SOX-2, which could play varying roles in tumorigenesis. For instance, prior research has shown that silencing SOX-2 promotes larger tumor development, whereas its overexpression correlates with smaller tumors, a finding consistent with our clonogenic assays showing smaller colonies at higher SOX-2 expression levels [56,57]. These observations underline the complexity of miR-486-5p's interaction with stemness markers such as OCT-4 and SOX-2, highlighting context-specific regulatory mechanisms. However, the exact pathways underlying these effects remain unclear, and further studies are necessary to validate and expand upon these findings, with a focus on the molecular mechanisms and their implications for therapeutic applications.
Reviewer 3 Report
Comments and Suggestions for Authors
The study investigates the role of miR-486-5p in colorectal cancer (CRC), focusing on its influence on cancer stem cells, a subpopulation linked to metastasis and treatment resistance. Etzi et al. found that miR-486-5p is significantly downregulated in CSC-enriched models, and its overexpression reduces CSC colony size, highlighting its onco-suppressive function. These findings are claimed to deepen the understanding of miR-486-5p role in CSC phenotype regulation and its potential as a therapeutic target in CRC. The article is written carefully and in clear language, and I did not notice any linguistic errors. The research methods used are described clearly, and the description is complete to the best of my knowledge. However, I have a few comments and questions for the authors, which are outlined below.
- Line 104-106: "MicroRNAs (miRNAs) are endogenous short non-coding RNA sequences of 18-25 nucleotides that regulate gene expression at the post-transcriptional level in a sequence-specific manner [15]. Due to their high stability and the possibility of detecting them in human body fluids, miRNAs are being studied as a new class of valuable biomarkers" - I kindly suggest citing some more papers on studies concerning miRNA biomarkers, also those related to cancer research (for example, Zhao et al. 2019, doi: 10.1016/j.biopha.2019.108947; Lehmann et al. 2020, doi:10.3390/app10207275).
- In the case of statistical analysis, I suggest calculating the median (instead of or in addition to the mean). The advantage of the median over the mean lies in its resistance to the influence of outliers and skewed data distributions. Regarding this I have additional questions: were there any significant outliers in the data? was the distribution skewed?
- Lines 334-340: The fold change for Vimentin in HT-29 CSCs is remarkably high (71.30). Can you comment on potential technical or biological factors contributing to this result?
- Fig 1, Fig 2/Fig 3, Fig 5, etc.: the same colors are used to represent different things. It is misleading, and although the legend explains its meaning, I discourage using the same color to represent completely different things.
Author Response
Comment 1. Line 104-106: "MicroRNAs (miRNAs) are endogenous short non-coding RNA sequences of 18-25 nucleotides that regulate gene expression at the post-transcriptional level in a sequence-specific manner [15]. Due to their high stability and the possibility of detecting them in human body fluids, miRNAs are being studied as a new class of valuable biomarkers" - I kindly suggest citing some more papers on studies concerning miRNA biomarkers, also those related to cancer research (for example, Zhao et al. 2019, doi: 10.1016/j.biopha.2019.108947; Lehmann et al. 2020, doi:10.3390/app10207275).
Answer 1.
Authors: We would like to thank the reviewer for the suggestion, the appreciation is very interesting and appropriate. We agree, and we have edited the manuscript according to his suggestions as shown below.
Line 107-115:
Due to their high stability and the possibility of detecting them in human body fluids, miRNAs are being studied as a new class of valuable biomarkers. Numerous studies have demonstrated their utility in biomedical research, particularly in cancer [16]. For instance, miR-128-3p has shown a key role in suppressing breast cancer cellular progression by targeting LIMK1, highlighting its potential as a biomarker and therapeutic target in this disease [17] (Zhao et al., 2019, doi: 10.1016/j.biopha.2019.108947). Similarly, the rs12976445 polymorphism in miR-125a has been identified as a factor altering its expression, influencing breast cancer susceptibility and progression, further reinforcing the importance of miRNAs as molecular markers [18] (Lehmann et al., 2020, doi: 10.3390/app10207275) ….
Comment 2. In the case of statistical analysis, I suggest calculating the median (instead of or in addition to the mean). The advantage of the median over the mean lies in its resistance to the influence of outliers and skewed data distributions. Regarding this I have additional questions: were there any significant outliers in the data? was the distribution skewed?
Answer 2.
Authors: Dear reviewer, thank you for your insightful feedback. We opted to use the mean instead of the median in presenting our qPCR data because the normalization process intrinsic to qPCR (e.g., ΔCt or ΔΔCt methods) reduces variability and outliers, making the mean a robust measure of central tendency for these data. Regarding the colony formation assay, we chose to use the mean as the central measure for our analysis. This decision was based on the structure of the experiment, where colonies were categorized into predefined size ranges (e.g., 2,500–5,000 μm²). These ranges inherently exclude outliers and ensure that extreme values do not unduly influence the results.
Comment 3. Lines 334-340: The fold change for Vimentin in HT-29 CSCs is remarkably high (71.30). Can you comment on potential technical or biological factors contributing to this result?
Answer 3.
Authors: Following the reviewer's suggestion, we have modified text in the discussion section adding more relevant and actualized references about the role of vimentin in CSCs.
Line 653-664:
Due to their role in epithelial-to-mesenchymal transition (EMT), miRNAs are considered markers of this process [69-71]. When assessing the expression levels of EMT biomarkers in colonospheres obtained from HT-29 and T84 cell lines, the data showed that Vimentin was overexpressed in HT-29 colonospheres compared to the adherent counterpart, while SLUG and Vimentin were overexpressed in T84 colonospheres compared to T84 adherent cells. Vimentin is not only involved in EMT but also associated with CSCs, promoting resistance to apoptosis and tumor invasion, key elements in metastatic progression. Moreover, vimentin regulates CSC responses to fractionated radiation exposure in cervical cancer, highlighting its role in the survival and proliferation of these stem cells [72] [https://doi.org/10.3390/ijms24043271]. In colorectal cancer, vimentin-positive circulating tumor cells have shown to be a prognostic biomarker for advanced stages of the disease [73] [https://doi.org/10.1038/s41598-023-45951-1]
Comment 4. Fig 1, Fig 2/Fig 3, Fig 5, etc.: the same colors are used to represent different things. It is misleading, and although the legend explains its meaning, I discourage using the same color to represent completely different things.
Answer 4.
Authors: Following the reviewer's suggestion, we have modified the colors of the graphs.
Round 2
Reviewer 1 Report
Comments and Suggestions for Authors
accepted without further revision
Author Response
We sincerely thank you for your valuable time and insightful suggestions, which have significantly contributed to improving the quality of our work.
Reviewer 2 Report
Comments and Suggestions for Authors
Authors have addressed all my previous points.
It is advised to improve figuer quality by the following
1. Add sample size (N) to each bar graph;
2. In Figure 7f and 8f, the last item on x-axis can be ">20,100 μm2";
3. In Figure 7g and 8g, the current presentation of colony area is confusing (e.g. not showing an error bar). It should be a bar graph as well.
Author Response
Answers to reviewer 2
It is advised to improve figure quality by the following
Reviewer Q1. Add sample size (N) to each bar graph;
Author A1: Thank you very much for your valuable suggestion. We truly appreciate your feedback and have incorporated the sample size (N) into each bar graph, as well as in the respective figure legends.
Reviewer Q2. In Figure 7f and 8f, the last item on x-axis can be ">20,100 μm2";
Author A2: Thank you for your suggestion. We have updated Figures 7f and 8f to reflect the change, and the last item on the x-axis now reads ">20,100 μm²" as recommended.
Reviewer Q3. In Figure 7g and 8g, the current presentation of colony area is confusing (e.g. not showing an error bar). It should be a bar graph as well.
Author A3: Thank you very much for your valuable suggestion. We sincerely appreciate your thoughtful input. We attempted to implement the suggested modification; however, we encountered significant challenges in merging the two datasets into a single bar graph.
Specifically, the numerical differences between the two datasets make it impractical to display them together on the same scale without compromising clarity. Additionally, the use of different units of measurement further complicates their simultaneous representation within the same diagram.
To provide a clearer perspective, we have attached the relevant images, which illustrate the challenges we faced and demonstrate the reasoning behind our decision to maintain the previous graph.
Thank you once again for your time and thoughtful review.